# IL-33 Enhances ACE2 Expression on Epidermal Keratinocytes in Atopic Dermatitis: A Plausible Issue for SARS-CoV-2 Transmission in Inflamed Atopic Skin

**DOI:** 10.3390/biomedicines10051183

**Published:** 2022-05-20

**Authors:** En-Cheng Lin, Chien-Hui Hong

**Affiliations:** 1Department of Dermatology, Kaohsiung Veterans General Hospital, Kaohsiung 813414, Taiwan; eclin@vghks.gov.tw; 2Department of Dermatology, School of Medicine, National Yang Ming Chiao Tung University, Taipei 11221, Taiwan

**Keywords:** interleukin-33 (IL-33), angiotensin converting enzyme 2 (ACE2), atopic dermatitis, COVID-19, SARS-CoV-2

## Abstract

Background: Interleukin-33 (IL-33) is an important cytokine in the pathophysiology of atopic dermatitis (AD) and in the progression of COVID-19. Angiotensin converting enzyme 2 (ACE2), the entry receptor for SARS-CoV-2, is expressed in epidermal keratinocytes. Whether IL-33 could regulate the expression of ACE2 mechanistically in keratinocytes warrants investigation. Objective: We questioned whether the ACE2 expression is increased in AD skin. We also questioned whether ACE2 is expressed in keratinocytes; if so, would its expression be enhanced mechanistically by IL-33. Methods: We measured and compared the expression of ACE2 in skin from patients with AD, patients with psoriasis, and healthy controls using immunohistochemistry. Flow cytometry, immunofluorescent exam, and quantitative RT-PCR were used for measuring the ACE2 expression in cultured keratinocytes treated with IL-33 and IL-17. Blocking antibodies were utilized to study the intracellular signaling pathways governing the ACE2 expression using cytokines. Results: The results showed that the ACE2 expression is increased in AD compared with that in healthy skin and psoriasis. In primary epidermal keratinocytes, ACE2 is constitutively expressed. IL-33 induces a time-dependent increase in ACE2 expression in cultured keratinocytes through quantitative PCR, flow cytometry, and immunofluorescent examinations. Furthermore, pretreatment of an ERK inhibitor, but not a STAT3 inhibitor, eliminated the increases in ACE2 by IL-33 in keratinocytes, indicating that IL-33 enhances ACE2 expression through ERK on epidermal keratinocytes. Conclusion: This is the first study to reveal that IL-33 enhances ACE2 expression on keratinocytes via ERK. Although further mechanistic studies are required, the increased ACE2 expression in IL-33 might have a biological implication on the transmission of SARS-CoV-2 in patients with AD.

## 1. Introduction

Atopic dermatitis (AD), a common inflammatory skin disease, affects 15–20% of children and 1–3% of adults in industrialized countries [1]. Intensive itching and sleep disturbance in AD cause huge burdens in patients, family, and society. Its pathogenesis includes impaired skin barriers and the aberrant production of T helper-2 (Th2)-associated cytokines [2]. When the epidermal barrier is impaired by exogenous insults or endogenous immune responses, IL-33 is released from epidermal keratinocytes as an alarmin to activate type-2 innate lymphoid cells (ILC2) [3] and basophils to release Th2 cytokines, including IL-4 and IL-13, which reduce the expression of barrier protein filaggrin, forming a viscous cycle to perturb the damaged skin barrier [4]. The IL-33 receptor, known as the membrane-bound stimulation-2 receptor (ST2), is widely expressed in several immune cells, such as ILC2, Th2 cells, regulatory T (Treg) cells, M2 macrophages, and eosinophils, along with epithelial cells, including lung epithelial cells [5] and skin epidermal keratinocytes [6]. Thus, during the process of the perturbed skin barrier, IL-33 is released from keratinocytes and acts on a variety of dermal and epidermal immunocompetent cells [7]. For example, IL-33 induces the release of IL-31 and histamine from Th2 cells [8] and mast cells, respectively [9]. Both IL-31 and histamine are mediators of pruritus [10]. In an animal model, Imai et al. revealed that transgenic mice (IL33tg) with keratin 14 promoter developed dermatitis and severe itching at the age of 5 weeks [11]. In addition, skin barrier proteins, including filaggrin and claudin-1, were decreased in IL33tg mice [12]. The important role of IL-33 in AD is further evidenced by a phase 2a clinical trial, demonstrating that Etokimab, an anti-IL-33 monoclonal IgG1 antibody, exhibited a therapeutic benefit for AD [13].

COVID-19 has caused a major pandemic throughout the last 2 years, and claimed millions of lives worldwide. Damaged alveolar epithelial cells are one of the major pathological findings in COVID-19 pneumonia [14]. IL-33 and other cytokines are released from these damaged epithelial cells. These cytokines have an impact on disease severity. In patients with mild symptoms, IL-33 stimulates Treg cells and ILC2 to release IL-4, which in turn promotes mast cells to release histamine, causing allergy-like symptoms in these patients [15,16]. In patients with mild to moderate symptoms, IL-33 induces ILC2, releasing a large amount of IL-9, expanding effector memory T cells in the lungs [17,18]. In patients with moderate to severe pneumonia, IL-33, IL-2, and IL-7 induce the expansion of ILC2, CD4 T cell, and GM-CSF-producing T cells [19,20]. In severe to critical cases, the cytokine storm is induced by IL-33, GM-CSF, and IL-1 [21,22]. These cytokines might provoke each other and act on multiple cell types, such as cardiac fibroblasts, endothelial cells, or adipocytes [23,24]. This phenomenon could cause multiple systemic diseases, such as thrombosis, heart failure, and even multiple organ failure [25,26,27]. In chronic stages, IL-33 stimulates M2 macrophage and ILC2 to release profibrotic cytokines, such as TGFβ and IL-13, involved in pulmonary fibrosis diseases [28,29]. Therefore, how IL-33 would regulate skin immune responses amid COVID-19 infection is interesting to know. On the other hand, IL-17, released from T-helper 17 cells (Th17), is involved not only in psoriasis, but also in the chronic phase of AD [30]. IL-17 could downregulate filaggrin or other skin barrier proteins [31,32,33]. We were also interested in discovering how IL-33 regulates skin immune responses amid COVID-19 infection.

SARS-CoV-2, the pathogen of COVID-19, infects lung cells via the angiotensin-converting enzyme 2 receptor (ACE2 receptor) [34]. ACE2 is an enzyme located on the cell membranes. The ACE2 transforms angiotensin II, a vasoconstrictor produced by angiotensin-converting enzyme (ACE), into the vasodilator angiotensin [35]. Similar to ST2, the ACE2 receptor is expressed not only in pulmonary type II alveolar cells, but also in other cells, including myocardial cells, proximal tubule cells of the kidney, ileum and esophagus epithelial cells, and bladder urothelial cells [36]. Zhu et al. demonstrated the relations between ACE2 expression on the keratinocytes and SARS-CoV-2 percutaneous transmission [37]. In human skin, Xue et al. showed ACE2 was expressed more on keratinocytes than that on other types of cells, such as fibroblasts and melanocytes. Through immunohistochemistry (IHC), ACE2-positive keratinocytes were enriched in the stratum basale, stratum spinosum, and stratum granulosum. This result suggests that skin could be a target organ for infection, particularly when the skin is inflamed with a perturbed skin barrier (as in the case of AD). Murine models showed that mRNA and the protein expression of ACE2 are induced by the application of imiquimod cream, MC903 (Calcipotriol, a vitamin D3 analog), and DNFB (1-Fluoro-2,4-dinitrobenzene), which mimic psoriasis, atopic dermatitis, and contact dermatitis, respectively [38]. Furthermore, one investigation in India revealed that the expression of angiotensin-converting enzyme 2 in psoriatic skin was enhanced by interferon-γ [39].

Therefore, IL-33 is one of the crucial cytokines in both atopic dermatitis and in COVID-19. We considered whether the ACE2 receptor expression in keratinocytes would increase in atopic dermatitis, and whether it would be attributed to the regulation and action of IL-33. In this study, we measured the expression of ACE2 in atopic skin and cultured keratinocytes, and investigated how IL-33 regulates ACE2 expression in epidermal keratinocytes.

## 2. Materials and Methods

### 2.1. Skin Samples for Immunohistochemistry

Human skin was obtained from AD through skin biopsy. All of the procedures were performed according to national and institutional ethical guidelines. Here, 5 µm serial tissue sections obtained from the skin of AD patients (age 20–64), patients with psoriasis (age 42–68), and controls (age 8–67) (n = 5, 4, and 9, respectively) were used.

### 2.2. Culture for Primary Keratinocytes

Normal human keratinocytes were obtained from adult foreskins through routine circumcision. The keratinocytes were harvested and cultured as described previously [40]. Briefly, skin specimens were washed with PBS (phosphate buffer saline) (pH 7.2), cut into small pieces, and harvested in Dulbecco’s Modified Eagle Medium (DMEM) containing 0.25% trypsin (Gibco, Grand Island, NY, USA) overnight at 4 °C. The epidermal sheet was separated from the dermis using fine-tipped forceps. The epidermal cells were pelleted through centrifugation (500× *g*, 10 min) and were then dispersed into individual cells by repeated gentle aspiration. The keratinocytes were gently resuspended in 5 mL of keratinocyte in a serum-free medium (Gibco), which contained 25 mg/mL bovine pituitary extract and 5 ng/mL recombinant human epidermal growth factor. Keratinocytes at the third passage were then grown in a keratinocyte in serum-free medium without bovine pituitary extract and recombinant human epidermal growth factor for 24 h before experimentation.

### 2.3. Immunohistochemistry

Tissues were fixed in 10% buffered formalin, dehydrated, and embedded in paraffin at the pathology department. Deparaffinization with xylene was performed in our lab. Subsequently, the sections were washed in PBS (phosphate buffer saline) and incubated with primary antibodies directed against ACE2 (Rabbit, Genetex GTX101395, San Antonio, TX, USA), CD207 (Langerin, Invitrogen 12-2075-82, Waltham, MA, USA), and isotype antibody (Rabbit IgG and Rat IgG2a-PE). The slides were then washed in PBST (phosphate buffer saline with 0.1% Tween 20) and visualization was performed using a fluorescence microscope (Olympus DP80, Shinjuku-ku, Tokyo, Japan) according to the manufacturer’s instructions.

### 2.4. Quantitative RT-PCR

The keratinocytes were cultured in a 12-well plate with keratinocyte SFM (serum-free medium). Cytokines were added to the medium with different concentrations and durations. The quantification and purity of the RNA were assessed using A260/A280 absorption (Nanodrop spectrophotometer; Thermo Fisher (Waltham, MA, USA)), and RNA samples with ratios greater than 1.7 were stored at −70 °C for further analysis. Extracted RNA (1 µL) was then subjected to PCR amplification using MPCR kits (Maxim Biotech, San Francisco, CA, USA) according to the manufacturer’s instructions. Primer sequences used were designed for human ACE2 FORWARD: CATTGGAGCAAGTGTTGGATCTT and human ACE2 BACKWARD: GAGCTAATGCATGCCATTCTCA. The reactions were carried out under the following conditions: 96 °C for 1 min and 60 °C for 4 min, 30 cycles of 94 °C for 1 min and 60 °C for 3 min, and extension at 70 °C for 10 min.

### 2.5. Flow Cytometry

We treated keratinocytes with IL-33 (100 ng/mL) and IL-17 (100 ng/mL) for 2, 6, and 24 h, respectively. The cells were detached from the cultured dishes by trypsin. The cells were fixed and holed by 1% Saponin for 30 min in room air. We stained the cells with 1:500 Rabbit (Genetex, GTX101395, San Antonio, TX, USA) overnight at 4 °C. Then, the cells were bound by Goat anti-rabbit IgG-568 antibody (secondary antibody) for 1 h in room air. We then measured the expressions of ACE2 by flow cytometry (BD Biosciences, San Jose, CA, USA).

The following cytokines and antibodies were used: IL-33: PeproTech #200-33, 100 ng/mL; IL-17: PeproTech #200-17, 100 ng/mL; ACE2: Rabbit, Genetex, GTX101395; Isotype: Rabbit IgG; secondary antibody: Goat anti-rabbit IgG-568.

In order to validate that IL-33 indeed enhances the expression of ACE2 in keratinocytes, we used an immunofluorescent examination to measure the expression of ACE2 in keratinocytes treated with IL-33. Furthermore, we were interested in the mechanism through which IL-33 enhances the expression of ACE2. Therefore, in order to test this hypothesis, inhibitors of ERK (PD98059) or STAT3 (STA21) were added to the keratinocytes treated with IL-33.

### 2.6. Immunofluorescent Exam

Keratinocytes were cultured with keratinocyte SFM (serum-free medium). KCs, attached on 16 mm cover glasses, were cultured in 12-well plates. Cytokines and protein inhibitors were added to the 12-well plate for 24 h. Then, the cells were fixed and holed by 0.1% triton X for 1 h in room air. We stained the keratinocytes with 1:500 rabbit anti-ACE2 (Genetex GTX101395) overnight at 4 °C, followed by Goat anti-rabbit IgG-568 antibody (secondary antibody) binding for 1 h in room air. Immunostained samples were analyzed using the feature of “Analyze Particles” in ImageJ.

The following cytokines and protein inhibitors were used: IL-33: PeproTech #200-17, 100 ng/mL; PD98059 (ERK 1/2 inhibitor): Sigma P215, 50 uM STA21 (STAT3 inhibitor): CAYMAN 14996, 250 nM. The following antibodies were used for immunofluorescent staining: Isotype IgG: Rabbit IgG, ACE2: Rabbit, Genetex GTX101395. 4,6-Diamidino-2-phenylindole (DAPI) (Fluka) was used for the staining of the nuclei.

## 3. Results

### 3.1. ACE2 Expression Was Increased in Atopic Dermatitis Compared with That in Normal Skin

In order to investigate whether the ACE2 receptor was indeed expressed on the skin in vivo, we took skin samples from AD patients and psoriasis patients and performed an immunohistochemical analysis (Figure 1). The results showed that ACE2 was expressed in the basal and suprabasal epidermis. Of note, the ACE2 expression was significantly increased in AD compared with that in normal skin and psoriasis.

### 3.2. Through Quantitative RT-PCR, ACE2 mRNA Expression Increased in a Time-Dependent Manner under IL-33 Stimulation

We demonstrated the expression of ACE2 in the epidermal keratinocytes in tissue using IHC. We then asked whether ACE2 was expressed in cultured keratinocytes, and if so, whether IL-33 would enhance ACE2 expression. To address this, we performed quantitative RT-PCR to measure the transcriptional expression of ACE2 in keratinocytes treated with IL-33 for 2, 6, and 24 h. The result revealed that the ACE2 mRNA expression is present in keratinocytes (Figure 2). Under IL-33 stimulation, the ACE2 transcriptional expression was enhanced over time (24 h > 6 h > 2 h). However, there was no dose-dependent effect of IL-33 on the expression of ACE2. In contrast with that of IL-33, the ACE2 expression was not enhanced by IL-17, a cytokine closely relevant to the pathophysiology of psoriasis.

### 3.3. ACE2 Is Constitutively Expressed in the Cultured Keratinocytes — Both IL-33 and IL-17 Enhance the ACE2 Protein Expression in a Time-Dependent Manner

We showed that IL-33 enhanced the ACE2 transcriptional expression in keratinocytes. We then asked whether this would be verified at the translational level. We first used flow cytometry to measure the ACE2 expression in cultured keratinocytes (Figure 3). The results showed that the treatment of IL-33 or IL-17 at 10 ng/mL both enhanced the expression of ACE2 in cultured keratinocytes in a time-dependent manner.

### 3.4. IL-33 Induces the ACE2 Expression, Which Is Abrogated by Pretreatment with PD98059

We then used an immunofluorescent examination to measure the expression of ACE2 in keratinocytes treated with IL-33. The results showed that the ACE2 expression increased on the keratinocytes and IL-33 enhanced the expression of ACE2 on keratinocytes. In order to reveal the probable mechanism, inhibitors for ERK (PD98059) or STAT3 (STA21) were added to keratinocytes treated with IL-33. The expression of ACE2 was measured by immunofluorescent examinations and flow cytometry (Figure 4 and Figure 5, respectively). The immunofluorescent examination results showed that IL-33 consistently enhanced the expression of ACE2. Of note, pretreatment of the keratinocytes with PD98059, but not STA21, eliminated the increase in ACE2 expression by IL-33. Through flow cytometry, the data showed that IL-33 induced a modest expression of ACE2, which was abrogated by both PD98059 and STA21. Interestingly, while IL-17 induced a minimally increased expression of ACE2, the pretreatment of STA21 potentiated the expression of ACE2 by IL-17. Taken together, both immunofluorescent examination and flow cytometry data indicated that IL-33 induces the expression of ACE2 through ERK.

## 4. Discussion

In this study, we demonstrated that the ACE2 expression was increased in AD skin, but not in psoriatic skin. We also showed that the ACE2 expression in keratinocytes was enhanced in a time-dependent manner by IL-33 at both a transcriptional level and a translational level. In fact, the induction of ACE2 by IL-33 was mediated by ERK.

In our IHC (Figure 1), an increased ACE2 expression was presented in AD, but not in psoriasis. This result was consistent with the results in Xue et al.’s research [38]. In quantitative RT-PCR (Figure 2), we showed the time-dependent expression of ACE2 after IL-33 treatment at a transcriptional level. However, the enhancement of ACE2 by IL-33 was not dose dependent. At a translational level, we utilized flow cytometry to measure the ACE2 protein expression (Figure 3). Under IL-33 stimulation, the ACE2 protein expression increased in a time-dependent manner. To summarize, IL-33 enhanced the ACE2 expression at a transcriptional and translational level. Incidentally, it was paradoxical that IL-17 enhanced the ACE2 expression at a translational level, but not a transcriptional level. It is necessary to conduct more research to reveal the impact of IL-17 on the ACE2 expression.

We demonstrated that IL-33 induced the expression of ACE2 via ERK. In 2016, Ryu et al. showed that IL-33 downregulated the filaggrin expression by inducing signal transducer and activator of transcription 3 (STAT3) and extracellular signal-regulated protein kinase (ERK) phosphorylation in human keratinocytes [41]. They also demonstrated that IL-33 downregulated the expression of CLDN1, a tight junction protein on keratinocytes, via the phosphorylation of the ERK/STAT3 pathway [42]. Extracellular signal-regulated protein kinase (ERK)1/2 is a mitogen-activated protein kinase (MAPK) family protein with typical cascade signaling characteristics and plays an important role in signal transduction pathways and the function of transcription factors, including activator protein-1, proto-oncogene c-Fos, and ETS domain-containing protein Elk-1 [43]. In brief, ERK 1/2 has a role in delivering extracellular signals to the nucleus, and these signals regulate the cell cycle, cell proliferation, and cell development [44]. Signal transducer and activator of transcription (STAT) is a class of transcription factors that are activated by cytokines, growth factors, and other peptide ligands [45]. In humans, the Stat family consists of seven proteins, including STAT-1, -2, -3, -4, -5A, -5B, and -6 [46]. STAT-3 regulates a variety of functions, including proliferation, cell cycle progression, apoptosis, angiogenesis, and immune evasion [47,48,49]. In AD, the activation of the IL-13/IL-4–JAK–STAT6/STAT3 axis downregulates the expression of filaggrin, loricrin, and involucrin [50]. The dysfunction of these proteins impairs the skin barrier. On the other hand, IL-17 reduces the expression of filaggrin and involucrin via P38/ERK MAPK signaling pathways [51].

Our data showed that the IL-17 expression increased through IL-17 at a protein level, but was not upregulated at an mRNA level. Theoretically, the increased ACE2 protein level on the cell membrane of keratinocytes through IL-17 could result from ACE2 protein trafficking, but not the increased ACE2 expression through the new protein synthesis by transcription and translation. However, the mechanisms for the discrepancy in the transcriptional and translational levels of ACE2 through IL-33 and IL-17 remain unknown. A paper by Dr. Krueger showed that secukinumab, an IL-17 inhibitor, lowers the ACE2 expression in psoriatic skin [52]. Interestingly, in that report, the reduction in ACE2 at a protein level by tissue immunohistochemistry seemed to be more prominent than that at a transcriptional level by real-time tissue PCR.

One of the important limitations of this study was that we only had the pharmacological inhibition, but not RNA interference examination, to demonstrate the regulation of ERK in IL-33-induced ACE2 expression. Additional RNA interference experiments targeting ERK and/or other intracellular signals might provide further evidence. Nevertheless, we found scientific evidence that the ACE2 expression increased in atopic skin and epidermal keratinocytes treated with IL-33.

## 5. Conclusions

The ACE2 receptor is expressed not only in pulmonary cells, but also in keratinocytes. The expression of ACE2 is increased in the epidermis in AD. IL-33 increases the ACE2 expression on keratinocytes through ERK. Although SARS-CoV-2 has an affinity to pulmonary cells through ACE2, the expression of ACE2 in epidermal keratinocytes cannot be overlooked. Although further mechanistic studies are required, the increased ACE2 expression in IL-33 might have a biological implication on the transmission of SARS-CoV-2 in patients with AD.

## Figures and Tables

**Figure 1 biomedicines-10-01183-f001:**
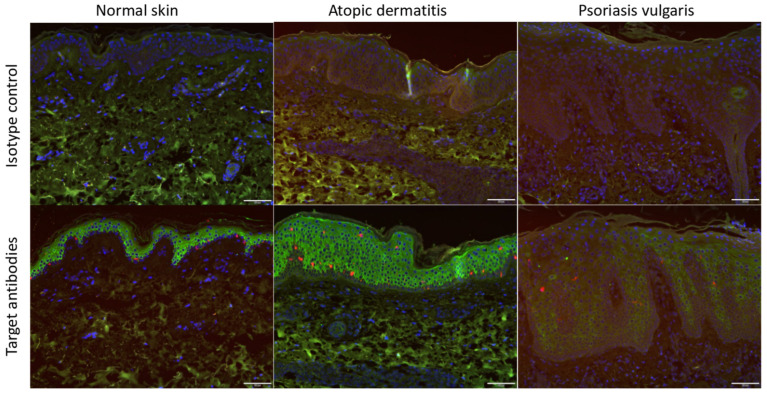
The ACE expression is increased in AD skin compared with that in psoriatic skin and healthy skin. We performed immunohistochemistry on the skin from psoriatic, atopic, and healthy patients. The ACE2 receptor (green color) was expressed in the basal and suprabasal epidermis in all three disease conditions. Notably, the ACE2 expression was increased in AD (**middle** pictures) compared with that in normal skin (**left**) and psoriasis (**right**). Green: ACE2; blue: DAPI; red: CD207 (langerin), Langerhans cell. Representative images are shown.

**Figure 2 biomedicines-10-01183-f002:**
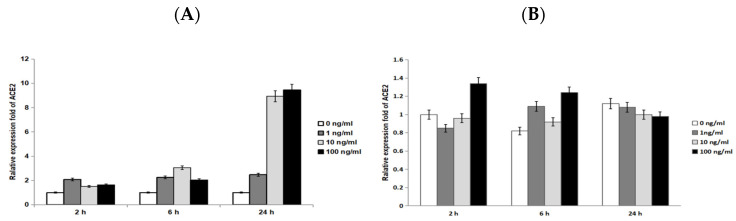
IL-33 enhanced the mRNA expression of ACE2 in a time-dependent manner. We performed quantitative RT-PCR for measuring ACE2 mRNA expression in keratinocytes treated with IL-33 (**A**) or IL-17 (**B**) at indicated time points. The results showed that IL-33 enhanced the mRNA expression of ACE2 in a time-dependent manner, but not in a dose-dependent manner (**A**). On the other hand, IL-17 did not enhance the ACE2 mRNA expression (**B**).

**Figure 3 biomedicines-10-01183-f003:**
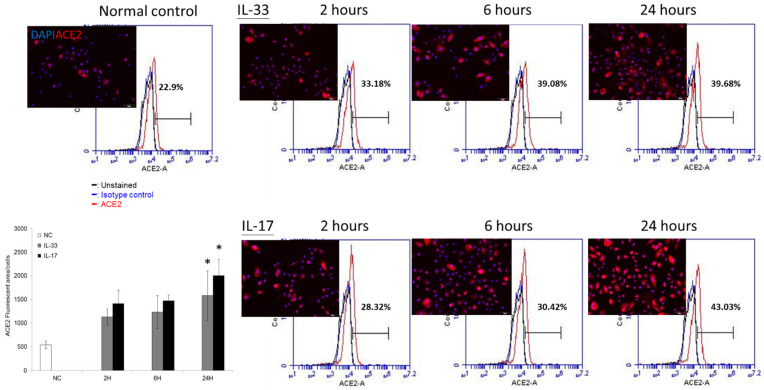
ACE2 is constitutively expressed in the cultured keratinocytes. We performed flow cytometry and immunofluorescence examinations for evaluating the ACE2 expression. Keratinocytes were treated with IL-33 and IL-17 at 10 ng/mL for 2, 6, and 24 h, respectively. The ACE2 protein expression under IL-33 (upper row) or IL-17 stimulation (lower row) was measured using flow cytometry and immunofluorescence examinations (representative data from three repeated experiments). The bar graph represents the quantification of the intensity of the ACE2 expression through immunofluorescence examinations. * indicates *p* < 0.05 compared with baseline ACE2.

**Figure 4 biomedicines-10-01183-f004:**
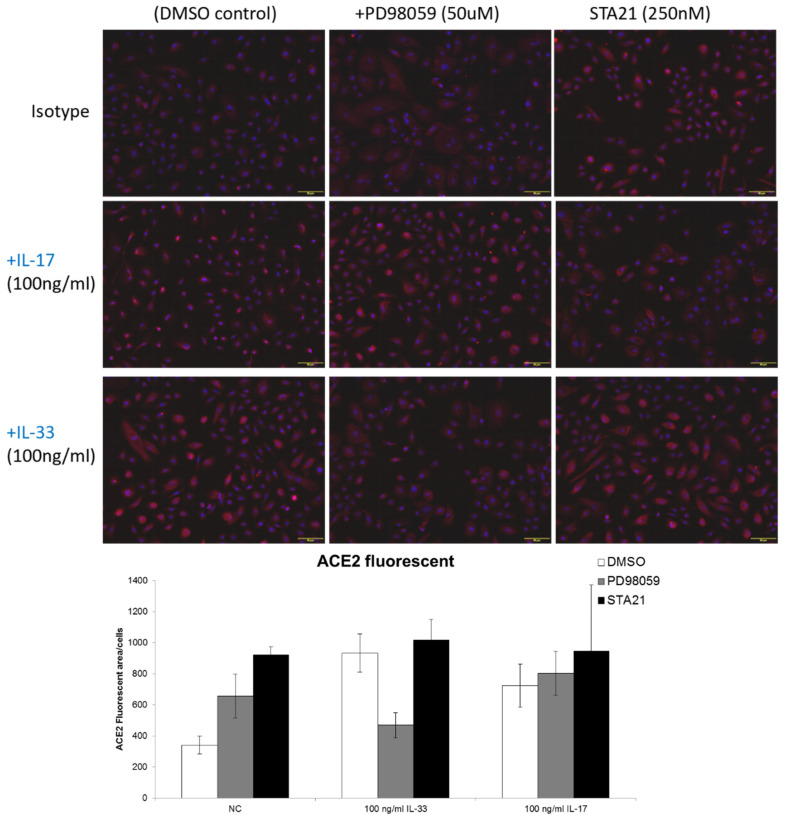
IL-33 enhanced ACE2 expression through ERK in keratinocytes. We performed immunofluorescent staining to measure the ACE2 expression in IL-33-treated keratinocytes. Small molecule inhibitors for PD98059 or STA21 (ERK and STAT3, respectively) were also pretreated. In the DMSO control, the fluorescence intensity of ACE2 (red color) was enhanced by IL-33 (**bottom left**). The fluorescence intensity decreased after PD98059 pretreatment (**bottom middle**). This phenomenon was not revealed after STA21 pretreatment (**bottom right**). Red: ACE2; blue: DAPI. The bar graph shows the quantitative data. In the DMSO control (white bar), the value of fluorescent area per cell increased under IL-33 stimulation. Moreover, under IL-33 stimulation (middle histogram), the value dramatically decreased after PD98059 pretreatment (gray bar), but not after STA21 pretreatment (black bar).

**Figure 5 biomedicines-10-01183-f005:**
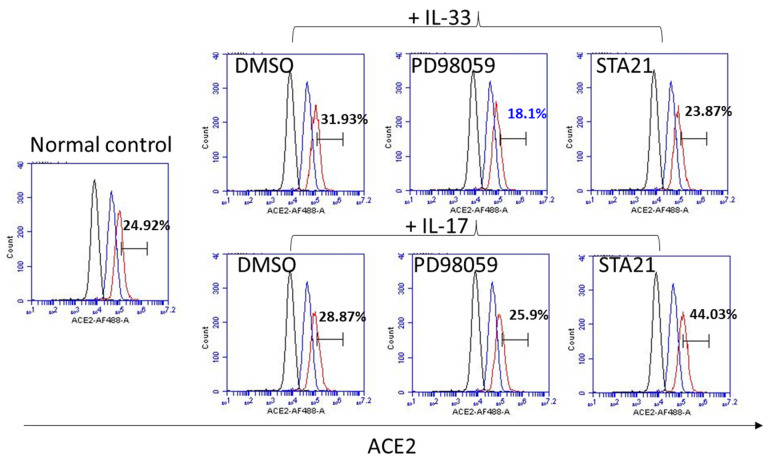
IL-33 enhanced ACE2 expression through ERK in keratinocytes by flow cytometry. With a similar experimental design, we performed flow cytometry to measure ACE2 expression in IL-33-treated keratinocytes. Small molecule inhibitors for PD98059 or STA21 (ERK and STAT3, respectively) were pretreated in order to investigate the role of ERK or STAT3 in IL-33-induced ACE2 expression. The data showed that IL-33 induced a modest expression of ACE2, which was abrogated by both PD98059 and STA21. Interestingly, while IL-17 induced a minimally increased expression of ACE2, the pretreatment of STA21 potentiated the expression of ACE2 by IL-17.

## Data Availability

Data are available from Kaohsiung Veterans General Hospital. Further inquiries can be directed to the corresponding author.

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
