# Peer review of "IL-33 Enhances ACE2 Expression on Epidermal Keratinocytes in Atopic Dermatitis: A Plausible Issue for SARS-CoV-2 Transmission in Inflamed Atopic Skin"

_biomedicines, 2022, doi:10.3390/biomedicines10051183_

Round 1

Reviewer 1 Report

In this manuscript, entitled "IL-33 enhances ACE2 expression through ERK on epidermal 2 keratinocytes in atopic dermatitis: a plausible issue for the 3 SARS-CoV2 transmission in inflamed atopic skin" the authors propose that IL33 enhances ACE2 expression through ERK. The experiment showing that ERK, and not STAT, is responsible for the observed effects, is weak. The experiment must be repeated with siRNA/shRNA or other knockout cell models. Then the protein should be reconstituted in the knockout or knockdown cell line to prove that that is the pathway through which the observed effects are  mediated. The current mechanism is weak and there lacks substantial evidence. In addition, the quality of the figures is quite poor. The immunofluorescence images should be replaced with better quality images. The COVID link is an exaggeration in the current form and lacks evidence. The manuscript needs further work,

Author Response

Reviewer 1

Comments and Suggestions for Authors

In this manuscript, entitled "IL-33 enhances ACE2 expression through ERK on epidermal 2 keratinocytes in atopic dermatitis: a plausible issue for the 3 SARS-CoV2 transmission in inflamed atopic skin" the authors propose that IL33 enhances ACE2 expression through ERK. The experiment showing that ERK, and not STAT, is responsible for the observed effects, is weak. The experiment must be repeated with siRNA/shRNA or other knockout cell models. Then the protein should be reconstituted in the knockout or knockdown cell line to prove that that is the pathway through which the observed effects are mediated. The current mechanism is weak and there lacks substantial evidence. In addition, the quality of the figures is quite poor. The immunofluorescence images should be replaced with better quality images. The COVID link is an exaggeration in the current form and lacks evidence. The manuscript needs further work

Response:

Thanks for the insightful comments. We agree that other than current pharmaceutical inhibitor approach, interference RNA approaches coupled with protein replacement would provide further evidences of the ERK involvement in the IL-33-induced ACE2 expression. However, with the limited time frame for the manuscript revision, we decided to use different approaches to measure ACE2 expression after IL-33 treatment, by flow cytometry and by immunofluorescent exam. The inhibition of IL-33 induced ACE2 expression was abrogated not only by our original experiments using immunofluorescent exam (Figure 4) but also by new experiments using flow cytometry (updated Figure 5). Other than cell experiments, we believe that the differential tissue expression of ACE2 in atopic dermatitis and psoriasis vulgaris by immunohistochemistry warrants reporting (Figure 1). The figure quality has been improved with better resolution embedded in the Microsoft Word file. We revise the conclusion by stating that

“Although further mechanistic studies are required, the increased ACE2 expression in IL-33 might have a biological implication on the transmission of SARS-CoV2 in patients with AD.”

Reviewer 2 Report

In this article, authors showed that ACE2 expression is increased in AD than that in healthy skin and psoriasis. In primary epidermal keratinocytes, ACE2 is constitutively expressed. IL-33 induces a time-dependent increase of ACE2 expression in cultured keratinocytes by quantitative PCR, flow cytometry and immunofluorescent exam. Furthermore, pretreatment of an ERK inhibitor, but not a STAT3 inhibitor, obliterated the increases of ACE2 by IL-33 in keratinocytes, indicating that IL-33 enhances ACE2 expression through ERK on epidermal keratinocytes. This is the first study to reveal that IL-27 enhances ACE2 expression on keratinocytes via ERK. The increased ACE2 expression in IL-33 might have a biological implication on the transmission of SARS-CoV2 in patients with AD. The results obtained are new and very informative data. I have some questions.

1) In Figure 2 and 3, ACE2 expression was elevated by IL-33 in mRNA and FACS expression. On the other hand, ACE2 expression was not upregulated by IL-17 at the mRNA level, but only by FACS. What is the possible mechanism for this phenomenon? If it is thought that IL-33 regulates ACE2 at the gene level and IL-17 upregulates ACE2 expression at the protein level, are there any previous reports on any possible mechanisms?

2) In Figure 4, the experimental results suggest that the increased expression of ACE2 may be mediated by ERK and not by STAT. However, the experimental results shown are only immunostaining, which is considered unreliable; similar inhibitor experiments with RT-PCR or FACS should be performed.

3) The text is colloquial and needs to be reworded to be appropriate for the paper. 

Altogether, the message that AD needs to be controlled to prevent COVID-19 is a very valuable clinical message. I believe that the data obtained is also valuable and should be read by many people.

Reviewer 3 Report

Reviewers

General comment-This is a clearly presented and well-written paper. In this manuscript, the authors examined the angiotensin converting enzyme 2 (ACE2) expression status on epidermal keratinocytes in atopic dermatitis (AD). ACE2 (entry receptor for the SARS-CoV2) is expressed in various cells including keratinocytes. IL-33 is an important cytokine in the pathogenesis of AD and COVID-19 progression. This study demonstrates that ACE2 expression is increased in AD than in healthy skin and psoriasis, and IL-33 induces increase in ACE2 expression. The following study will help us to understand ACE2 regulation in keratinocytes and IL33 mediated ACE2 induction may have implication in the transmission of SARS-CoV2 in patients with AD.

Summary of the salient findings:

In primary keratinocytes, IL-33 induces ACE2 expression in time dependent fashion. Further, pretreatment of keratinocytes with ERK/STAT3 inhibitor, reveals that IL-33 induces ACE2 expression through ERK.

The proposed study is very interesting, but I have the following comments and concerns.

  1. This study revealed that ACE2 expression is upregulated in keratinocytes derived from AD. However, the mechanism of ACE2 regulation in keratinocytes is not clear,? How ERK regulates ACE2 and what transcription factors are involved?
  2. IL-33 induces ACE2 expression. Does allergens/histamine regulate ACE2 expression in primary keratinocytes?
  3. IL-33 regulated ACE2 expression both transcriptional and translational (RNA & protein) level whereas IL-17 at translation level. However, the changes are small with IL17 in early time intervals (Fig3, lower panel). On the other hand, there is no significant change in difference of ACE2 induction between 6, and 24h with IL-33 (Fig3, upper panel). It will be great if you show ACE2 induction by Western blotting or some other method.

 Minor comment

  1. There are no error bars in figure 2 (a, b panels) mRNA data
  2. Its not clear in text and figure 3, about the dose of cytokines used for time course experiment.
  3. References are numbered twice and in line 369-373 once.

I recommend the manuscript be accepted for publication, with addressing these concerns  

Round 2

Reviewer 1 Report

The quality of the experiments and the data are still quite poor and I do not recommend publication. I do not feel comfortable recommending publication without more rigorous molecular biology experiments to support the proposed hypothesis.

Author Response

Dear Editors and Reviewers,                                May 8 2022

    Thanks for the comments. Please see our response to the reviewer’s comments.

“The quality of the experiments and the data are still quite poor and I do not recommend publication. I do not feel comfortable recommending publication without more rigorous molecular biology experiments to support the proposed hypothesis.”

Response:

We have struggled to make revisions based on the last round of the comments from 3 reviewers. Additional cell experiments, including ACE2 quantitative expressions in IL-33 treated keratinocytes by both immunofluorescence exam and flow cytometry (Figure 3), have been performed. Moreover, the pharmacological inhibition of ERK by PD98059 consistently reduced the IL-33-induced ACE2 expression by immunofluorescence (Figure 4) and by flow cytometry (Figure 5). I understand that interference RNA experiments might provide additional evidences of how ERK regulates IL-33-induced ACE2 expression. With the short revision time (10 days), it remains difficult to establish the cell clone with stable transfection for this purpose. We discuss this limitation in the discussion section. In addition, we deleted the ERK regulation in the manuscript title as “IL-33 enhances ACE2 expression through ERK on epidermal keratinocytes in atopic dermatitis”.

“One of the important limitations of this study is that we only have the pharmaco-logical inhibition but not RNA interference exam to demonstrate the regulation of ERK in IL-33-induced ACE2 expression. Additional RNA interference experiments targeting ERK and/or other intracellular signals might provide further evidences. Nevertheless, we do have scientific evidences that ACE2 expression is increased in atopic skin and epidermal keratinocytes treated with IL-33.”

    I firmly believe that the enhanced ACE2 expression in atopic dermatitis and in IL-33-treated keratinocytes would be quite interesting to the readers of general interests with substantial biological importance. I sincerely hope the reviewer could consider these important findings again.

Best regards,

Chien-Hui Hong, MD, PhD

Department of Dermatology, Kaohsiung Veterans General Hospital, Kaohsiung, Taiwan

Emal: [email protected] ; Phone: +886-928329260

Round 3

Reviewer 1 Report

The authors have not addressed my concerns. The authors responded that 10 days was insufficient to perform the experiments. I believe that they can always request more time from the editor if they need to improve their science. The new flow cytometry and immunofluorescence experiments do not add anything to the manuscript and do not address my concerns. I do not recommend publication of this work.